# Could *Cratylia argentea* replace Tifton 85 hay on growing and finishing lamb diets in tropical areas?

Elaine Cristina Teixeira[1]*, Lucas Freires Abreu[2], Fernando Antônio de Souza[2], Walter José Rodrigues Matrangolo[3], Karina Toledo da Silva[4], Luciano Soares de Lima[2], Hemilly Cristina Menezes de Sa[2], Ângela Maria Quintão Lana[2]

1 Departamento de Engenharia Florestal, Universidade Federal de São João del Rei, Sete Lagoas, Minas Gerais, Brasil, 2 Departamento de Zootecnia, Escola de Veterinária, Universidade Federal de Minas Gerais, Belo Horizonte, Minas Gerais, 3 Empresa Brasileira de Pesquisa Agropecuária—Embrapa Milho e Sorgo, Sete Lagoas, Minas Gerais, Brasil, 4 Empresa de Pesquisa Agropecuária de Minas Gerais–EPAMIG Centro Oeste, Prudente de Morais, Minas Gerais, Brasil

* elaineteixeira@ufsj.edu.br

## Abstract

Legumes shrubs such as *Cratylia argentea* have an ability to thrive in environments with low water availability and poor soil. On the other hand, forage grasses such as Tifton 85 have a greater demand for inputs to be productive. The objective of this study was to evaluate the performance of growing and finishing Lacaune lambs fed *Cratylia argentea* hay as an alternative to Tifton 85 (*Cynodon* spp). Twenty-four Lacaune lambs aged between five and six months (average body weight [BW] 21.50 ± 3.38 kg) were arranged in a split-plot randomized block design. The plots consisted of different Cratylia to Tifton 85 hay proportions (0, 20%, 40%, or 100%, dry matter [DM] basis) as a roughage replacement in the total diet. The subplots represented two evaluation times, entitled "initial period" and "final period", which consisted of the early seven days of total feces and urine collection, and the last seven days of the experiment, respectively. The lambs were blocked by weight with six replicates per treatment. The results show that the level of Tifton 85 replacement for Cratylia hay in the roughage portion of the lamb diet did not influence ($P > 0.05$) weight gain (WG), dry matter intake or dry matter digestibility; feed conversion, feed efficiency; and the evaluated nitrogen balance variables. The digestibility coefficient of neutral detergent fiber decreased linearly as Tifton 85 replacement for Cratylia level was increased, which probably happened due to the presence of highly lignified material within the Cratylia hay. However, the alternative legume maintained animal performance of Tifton 85. In conclusion, Cratylia hay can be recommended as a potential substitute for Tifton 85 hay, which requires greater inputs for the production. Cratylia may be considered a feeding strategy for livestock production, especially for smallholder livestock systems and regions with unfavorable soil and climate.

**Data Availability Statement:** All relevant data are within the paper and its Supporting Information files.

**Funding:** This work was supported by the National Council for Scientific and Technological Development (CNPq, Brazil n° 140314/2019-9), State Agency for Research and Development of Minas Gerais (FAPEMIG, PPM-00639-18) and Coordination for the Improvement of Higher Education Personnel of Education (CAPES, Brazil n° 88887.667823/2022-00). The funders had no role in study design, data collection and analysis, decision to publish, or preparation of the manuscript.

**Competing interests:** The authors have declared that no competing interests exist.

# Introduction

Tifton 85 (*Cynodon* spp.) is one of the major grasses used for hay production in Brazil [1]. Nonetheless, tropical grasses such as Tifton 85 can have its feed value considerably reduced due to lower digestibility and quality over water deficit periods. Moreover, its yield and use are linked to the adoption of management practices associated with the intensification of inputs use such as chemical fertilization [2] and complementary irrigation.

The increased use of chemical fertilizers is linked to greater greenhouse gas (GHG) emissions and depletion of water resources, which happens due to nitrous oxide emissions and run-off, respectively [3]. In the same context, a rise in nitrogen fertilizer costs have led nitrogen (N) fixing crops, such as legume shrubs, to become more attractive for producers.

Several leguminous shrubs and trees have been identified as feed alternatives for ruminants in tropical areas [4,5]. However, its use is yet to be adopted, possibly due to a lack of consensus on the most suitable species and how to incorporate them into tropical production systems [4,6].

*Cratylia argentea* (Cratylia) is a perennial legume shrub with a feeding potential for livestock in tropical areas. Cratylia is very competitive not only in terms of mass production but also due to its great crude protein (CP) concentration largely because of its ability to symbiotically associate with nitrogen-fixing bacteria as well [7]. Cratylia can also maintain green leaves in critical periods of water deficit, thanks to its deep and spread root system [7]. In their initial growth phase, the Cratylia rootlets develop an external tissue similar to a cork, which helps to minimize water loss [7]. Those features allow Cratylia to keep growing during the dry season, despite being grazed or harvested [8]. The same authors reported that 30–40% of Cratylia's yield may occur in the dry season, which makes a great feedstock to fill forage gaps in tropical regions. Another important characteristic of this legume is its tolerance to acidic soils with high aluminum saturation and low natural fertility [9,10]. Dry biomass accumulation by this legume ranges from 14 to 21 t ha$^{-1}$ yr$^{-1}$, without fertilizer application or supplemental water, as reported in previous research [11].

Several studies have demonstrated the benefits of including legume shrubs as an alternative for diet and performance improvement on ruminants [12–14]. Those studies found that these plants constitute a path towards sustainability, whether through direct grazing or for the enrichment of diets supplied in a trough [4,6].

With that being said, it is assumed that Cratylia hay might be an alternative to hay of forage grasses in the diet of growing and finishing lambs. The objective of this study was to evaluate the performance of growing and finishing lambs fed diets with increasing levels of Cratylia hay to replace Tifton 85 hay as the roughage source.

# Materials and methods

## Study area

The study was conducted at the Brazilian Agricultural Research Corporation–Embrapa Milho e Sorgo (Sete Lagoas, Minas Gerais, Brazil; 19°28′ S; 44°15′W, at 732 m altitude), where the Cratylia was planted for hay, and the Laboratory of Animal Metabolism and Calorimetry (LAMA/LACA) at the Veterinary School of the Federal University of Minas Gerais (EV-UFMG), Belo Horizonte, MG, where the lambs and infrastructure for the feeding experiment were located.

The unit where Cratylia was planted is in a region with a Cwa climate type according to the Köppen classification system, i.e., savanna climate, with dry winters and humid-rainy summers. Cratylia was transplanted from a growth chamber to a research site on March 28th 2013,

**Table 1. Soil test result (00–20 cm) and planting date of *Cratylia argentea*.**

| Planting date | pH | NO$_3$-N | P | K | OM |
|---|---|---|---|---|---|
| | | ———————— (mg kg$^{-1}$) ———————— | | | (dag kg$^{-1}$) |
| 28/03/13 | 5.5 | 18 | 2.18 | 39.26 | 3.44 |

Soil pH was measured using a 1:1 soil:water, NO$_3$-N was measured using a 2M KCl extract, and plant available P and K were extracted using a Mehlich-3 solution [16].

and several trials were conducted in the area until 2019 when the current research began. Cratylia's shrubs were planted in an area of 105 m$^2$, spaced at 1 m between rows and a total of 126 plants, or 12,000 plants/ha$^{-1}$. The size of the research area was determined based on the amount of available Cratylia's sprigs found available with research peers before planting, since commercial seeds were not available at the time. The soil test for the area where Cratylia was planted for hay is described in Table 1. The Cratylia hay was produced entirely without the use of chemical fertilizers based on the potentially autogamous characteristic of the legume and to maintain the low production cost [15]. Tifton 85 hay was purchased from a local dealer and presented a dry matter (DM) of 91.25%, CP 13.2% and NDF 68.98%, which was used to formulate the feed mixture. The agronomical accessions for the harvested Cratylia hay are presented in Table 2.

## Cratylia hay and feed management

To establish a uniform regrowth, all Cratylia's plots were mowed in January 2019. The harvest for data collection happened on April 2$^{nd}$ of 2019, July 10$^{th}$, August 7$^{th}$ of 2019 and January 9$^{th}$ and April 14$^{th}$ of 2020. The stubble height was 25 cm [17], without the selection of less lignified material. During the haymaking period, temperatures were between 17.74 and 29.24°C, average rainfall was 885 mm, and relative humidity was 66.56% [18].

To produce enough forage mass for the totality of animals, after each harvest, the produced hay was stored in raffia bags and packed in metal barrels to avoid contamination, until the beginning of the feeding treatments at LAMA/LACA, which started in November 2019 and lasted until January 2020. All harvested material was cut through a forage chopper then spread in a 10-cm layer on a cemented area for drying in the sun and turned frequently for drying to the point of hay, which happened when the Cratylia reached a DM content of 85%. The interval of exposure to the sun for the material to reach the point of hay varied between 48 and 72 h, depending on the weather conditions. During the night, or in case of rain, the material was covered with canvas to prevent moisture.

The Tifton 85 and *Cratylia* hay were ground to a particle size of 5 mm, homogenized, and then mixed with the other ingredients of the experimental diets according to the replacement levels (Table 3), as described by [19].

**Table 2. Agronomical accessions for the harvested *Cratylia argentea* hay.**

| | DM | CP | ADF | ADIN | NDF | NDIN |
|---|---|---|---|---|---|---|
| | 93.2 | 19.6 | 41.6 | 0.492 | 58.4 | 2.28 |
| sd | 1.28 | 3.45 | 5.52 | 0.100 | 5.40 | 1.19 |

DM, dry matter; CP, crude protein; ADF, acid detergent fiber; ADIN, acid detergent insoluble nitrogen, NDF, neutral detergent fiber; NDIN, neutral detergent insoluble nitrogen; sd, standard deviation. $n$ = 530.

**Table 3. Percentage and chemical composition of experimental diets.**

| | Replacement of Tifton 85 with *C. argentea* | | | |
|---|---|---|---|---|
| **Variable** | **0%** | **20%** | **40%** | **100%** |
| Ingredient (g/kg DM) | | | | |
| *Cratylia argentea* hay | - | 100.0 | 200.0 | 441.3 |
| Tifton 85 hay | 549.9 | 425.4 | 300.7 | - |
| Ground maize | 237.2 | 266.4 | 295.6 | 366.1 |
| Soybean meal | 175.4 | 170.9 | 166.4 | 155.5 |
| Mineral for sheep[1] | 15.0 | 15.0 | 15.0 | 15.0 |
| Limestone | 10.5 | 10.5 | 10.8 | 11.3 |
| Dicalcium phosphate | 7.0 | 6.8 | 6.5 | 5.8 |
| Sodium bicarbonate | 5.0 | 5.0 | 5.0 | 5.0 |
| Chemical composition | | | | |
| Dry matter (g/kg) | 908.4 | 913.2 | 903.3 | 903.0 |
| Organic matter (g/kg DM) | 919.9 | 923.1 | 922.4 | 923.9 |
| Ash (g/kg DM) | 80.1 | 76.9 | 77.6 | 76.1 |
| Crude protein (g/kg DM) | 178.4 | 188.8 | 180.7 | 182.2 |
| Ether extract (g/kg DM) | 24.9 | 13.0 | 16.3 | 17.7 |
| Neutral detergent fiber (g/kg DM) | 433.4 | 426.0 | 390.1 | 352.5 |
| Total carbohydrates (g/kg DM) | 716.6 | 721.4 | 725.4 | 724.0 |

Tortuga® (DSM) Mineral for sheep (content per kg of product): zinc—3,800 mg; sodium—147 g; manganese—2,000 mg; cobalt—15 mg; copper—590 mg; sulfur—18 g; selenium—20 mg; iodine—50 mg; chromium—20 mg; molybdenum—300 mg; calcium—110 g; calcium (max.) - 135 g; fluorine (max.) - 870 mg; phosphorus—87 g. DM–dry matter, NM–natural matter.

## Feeding trial

Twenty-four Lacaune lambs aged between five and six months (21.5 ± 3.38 kg) were divided into groups according to their initial weight (Table 4). The lambs were housed in individual metabolic cages (1.5 × 0.7 m) equipped with a feeding and water trough, and a system for separate collection of feces and urine, inside a covered masonry shed with an exhaust system at LAMA/LACA laboratories.

The total experimental period was 64 days, the first 21 of which were used as a period of adaptation to the experimental diets and cage management, followed by 43 days of intake and weight gain (WG) measurements. Digestibility trials were conducted in two periods, here entitled "initial" and "final". The initial period consisted of 7 days of total collection of feces and urine, which began after the adaptation period (5 to 6 months, 26.8 ± 3.26 kg), and the final period began in the last 7 days of the experiment (6 to 7 months, 37.5 ± 3.26 kg).

Feed was supplied twice daily (08h00 and 14h00) for *ad libitum* intake (200 g/kg orts, as-fed basis). Supplied feed and orts were weighed daily on an electronic scale. Each of the diets was composed of different levels of Cratylia hay to replace Tifton 85 hay (Table 3). The diets were

**Table 4. Block distribution of lambs.**

| Block | Category | Mean weight (kg) | Standard deviation | Replicates per treatment |
|---|---|---|---|---|
| 1 | Light | 15.9 | 0.580 | 1 |
| 2 | Medium-light | 20.6 | 0.970 | 2 |
| 3 | Medium-heavy | 23.5 | 0.640 | 2 |
| 4 | Heavy | 26.0 | 1.21 | 1 |

formulated to meet the requirements for growing and finishing lambs with a DM intake of 4% BW and an average daily gain (ADG) of 200 g, thus, a 16% CP and total digestible nutrients (TDN) of 75%, following the [20]. All diets were supplied as a total mixture ratio and adjusted daily allowing up to 20% orts.

Live weight was measured weekly from the beginning of the experimental period, in the morning period before the feed was supplied, using a platform scale. Initial live weight (ILW) was determined immediately after the end of the adaptation period, and final live weight (FLW) at 43 days of the experiment. The ADG was determined by the following equation: ADG = ((FLW–ILW)/(Number of days in the feedlot). Dry matter intake (DMI) was individually calculated as the difference between the amount of feed supplied and daily orts.

The digestibility trial involved total collection of feces and urine on two occasions: (1) in the first week and (2) in the last week of the experimental period. Samples of supplied feed, orts, and feces were analyzed for the dry matter (DM), ash, and nitrogen (N) contents according to [21] and the neutral detergent fiber (NDF) content according to [22]. The total tract apparent digestibility coefficient (ADC) was determined as follows: ADC = ([OF–Intake–CF]/ OF–Intake) × 100, which was proposed by [23], where AD = apparent digestibility; OF = offered feed [(offered feed amount in kg DM) x (offered nutrient content in % DM)/ 100]; Intake = orts [(removed orts feed in kg DM) x (orts nutrient content in % DM)/100]; and CF = fecal output [(amount of collected feces in kg DM) x (collected feces content in % DM)/ 100].

Urine was collected in plastic containers containing a hydrochloric acid solution (6N HCl) to keep the urine pH below 2 and prevent the loss of nitrogen by volatilization. The total collected material was quantified, homogenized, and filtered through cotton gauze and a 10% sample was taken and stored at -18°C. The nitrogen content of the sample was determined by the micro Kjeldahl method [24]. Nitrogen balance was calculated according to [25].

## Statistical analyses

The 24 lambs were allocated in a randomized block design with four blocks, where each block was randomly assigned with a dietary treatment of 0, 20, 40, and 100% Cratylia hay replacing Tifton 85 hay in the diet, and six replicates. The effect of treatments was submitted to an analysis of variance (ANOVA), with the following general model:

$$Y_{ijkl} = \mu + \tau_i + \gamma_k + e_{ikl} + \beta_j + (\tau\beta)_{ij} + \epsilon_{ijkl},$$

where $Y_{ijkl}$ = value observed at the Tifton 85 replacement level for Cratylia hay level I, in block k, replicate l, and subplot j; $\mu$ = mean overall effect; $\tau_i$ = effect of Tifton 85 replacement level for Cratylia hay i; $\gamma_k$ = effect of block k; $\epsilon_{ikl}$ = error attributed to the plot $Y_{ikl}$ ($\epsilon_{ik} \sim N [\mu, \sigma_2]$); $\beta_j$ = effect of collection period j; $(\tau\beta)_{ij}$ = interaction between Tifton 85 replacement level for Cratylia hay i and collection period j; and $\epsilon_{ijkl}$ = error attributed to subplot $Y_{ijk}$ ($\epsilon_{ijkl} \sim N [\mu, \sigma_2]$).

For performance variables, the fits of polynomial regression models were tested to obtain the equation as a function of the Tifton 85 replacement levels for Cratylia hay or collection periods.

Analyses were carried out using R statistical analysis software [26]. For all analyses, the statistical assumptions of normality and homoscedasticity were evaluated by the Shapiro-Wilk and Bartlett tests, respectively. An effect was considered significant when the α error was less than 5%.

The experimental procedures were approved by the Ethics Committee on Animal Use (CEUA) at the Federal University of Minas Gerais (approval no. 192/2020).

## Results

### Intake and digestibility of Cratylia hay

There was no interaction between Tifton 85 hay replacement levels for Cratylia and evaluation period (P>0,05). In addition, no difference was observed on DM intake, no matter the replacement level or evaluation period (Table 5).

Dry matter intake did not differ (P>0.05) in response to Tifton 85 replacement levels for Cratylia (Table 5). However, there was an increase of 12.5%, 17.64%, 18.75% and 16.67% in DMI with the respective increase in the proportions (0%, 20%, 30%, 40% and 100%) of Cratylia in the final collection period (P<0.05), compared with the initial collection period. However, the nutrient digestibility was influenced by the replacement levels of Tifton 85 hay. The 0% and 20% replacement levels showed no difference for digestible OM and CP, and the 40% replacement level did not differ from 0, 20, and 100%. The 100% inclusion of Cratylia had the lowest digestibility for OM and CP. Moreover, the NDF digestibility decreased with higher inclusion levels of Cratylia hay.

As far as the digestibility coefficient of DM, no difference (P>0.05) was observed with Tifton 85 replacement levels for Cratylia hay or collection period. The digestibility coefficients of OM showed a significant (P<0.05) decrease from 715.1 to 681.7 g/kg DM, with increased proportions of Cratylia in the roughage portion of the diet to replace Tifton 85. The same trend was observed on CP, wherein greater proportions of Cratylia decreased (P<0.05) the CP digestibility from 743.8 to 710.9 g/kg DM, from control to the treatment with 100% of Cratylia hay. Regarding NDF, there was also a steady decrease (P<0.05) in digestibility, which ranged from 606 to 404.3 g/kg DM, with increased levels of Cratylia.

In respect to collection periods, the digestibility coefficient of OM had increased (P<0.05) from 712.2 to 727.2, 701.6 to 715.6, and 681.7 to 700.1 g/kg DM at 20%, 40%, and 100% from initial to the final collection period, respectively. The same was observed for NDF, which the digestibility increased (P<0.05) from 571.3 to 609.1, 512.9 to 538.8, 404.3 to 454.0 g/kg DM at 20%, 40%, and 100% from initial to the final collection period, respectively. Nevertheless, no significant differences between collection periods were observed on digestibility coefficients of DM and CP.

**Table 5. Intake and digestibility of dry matter and nutrients, feed conversion, and feed efficiency of feedlot Lacaune lambs fed diets containing different levels of *C. argentea* hay as a replacement Tifton 85 hay at two evaluation times.**

|  | Collection Period |  | SE | *P*-value | Level |  |  |  | SE | *P*-value |
|---|---|---|---|---|---|---|---|---|---|---|
|  | Initial | Final |  |  | 0% | 20% | 40% | 100% |  |  |
|  | Intake (kg/day) |  |  |  |  |  |  |  |  |  |
| **Dry matter** | 1.40b | 1.69a | 0.14 | <0.001 | 1.50 | 1.58 | 1.47 | 1.64 | 0.16 | 0.548 |
| **Organic matter** | 1.29b | 1.56a | 0.13 | <0.001 | 1.38 | 1.46 | 1.36 | 1.51 | 0.15 | 0.527 |
| **Crude protein** | 0.26b | 0.31a | 0.27 | <0.001 | 0.27 | 0.30 | 0.27 | 0.30 | 0.03 | 0.337 |
| **NDF** | 0.56b | 0.67a | 0.06 | <0.001 | 0.65 | 0.67 | 0.57 | 0.58 | 0.06 | 0.162 |
|  | Digestibility (g/kg DM) |  |  |  |  |  |  |  |  |  |
| **Dry matter** | 682.91 | 692.95 | 0.86 | 0.089 | 692.72 | 69754 | 688.64 | 672.84 | 0.72 | 0.055 |
| **Organic matter** | 702.64b | 714.78a | 0.62 | 0.030 | 715.62a | 719.72a | 708.60ab | 690.89b | 0.72 | 0.017 |
| **Crude protein** | 732.48 | 727.68 | 0.52 | 0.409 | 733.17ab | 747.73a | 727.27ab | 712.15b | 0.66 | 0.012 |
| **NDF** | 523.64b | 551.54a | 0.79 | 0.018 | 605.15a | 590.21a | 525.84b | 429.16c | 1.12 | <0.001 |
| **Feed conversion** | 5.78b | 6.92a | 0.31 | <0.001 | 6.11 | 6.38 | 6.29 | 6.62 | 0.45 | 0.893 |
| **Feed efficiency** | 0.17a | 0.14b | 0.01 | <0.001 | 0.17 | 0.16 | 0.15 | 0.15 | 0.01 | 0.748 |

Means followed by different letters on the same line differ statistically using the Tukey test (p<0.05); SE = standard error; L = level of Tifton 85 replacement for *C. argentea* hay in the roughage to replace Tifton 85 hay.

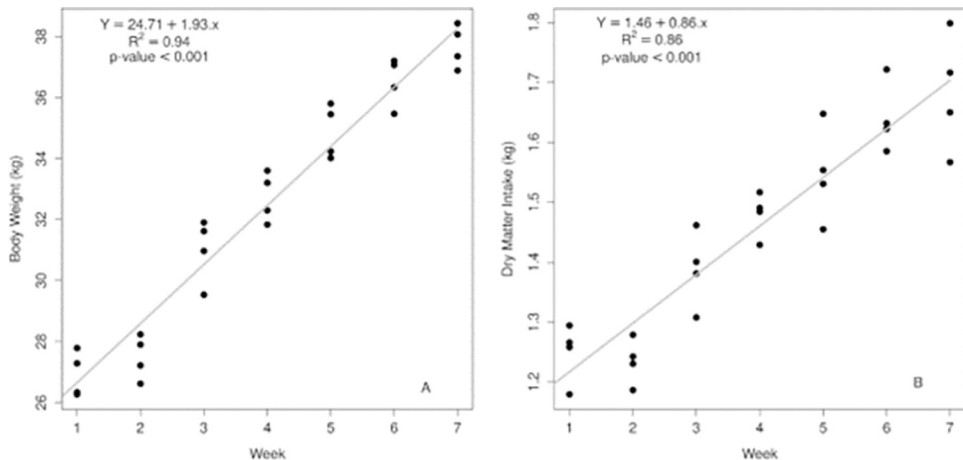

**Fig 1. Body Weight of lambs (A) as a function of weeks of hay feeding and Dry Matter Intake of the lambs (B) as a function of weeks of hay feeding.**

The feed conversion (FC) and feed efficiency (FE) did not differ (P>0.05) with Tifton 85 replacement levels for Cratylia's hay. However, FC was greater in the final period (P<0.05), while FE was greater in the initial period. It is important to note that, no matter the replacement levels or DMI parameters (OM, CP, NDF), the animals' weight gain met the expectations. There was no difference between replacement levels and animals' weight gain (p>0,05).

Dry matter intake was positively (P<0,01) influenced by weeks of hay feeding, whereas DMI was increased by 0.858 g per animal per week (Fig 1A). As far as WG, a similar pattern was observed, considering that WG increased 1.93 kg per animal per week (or 0.275 g/d) (P<0,01) (Fig 1B).

The digestibility coefficient of NDF decreased linearly by 0.18% every week with the Tifton 85 replacement for Cratylia hay (% DM) in the diet (Fig 2). Therefore, as estimated by the

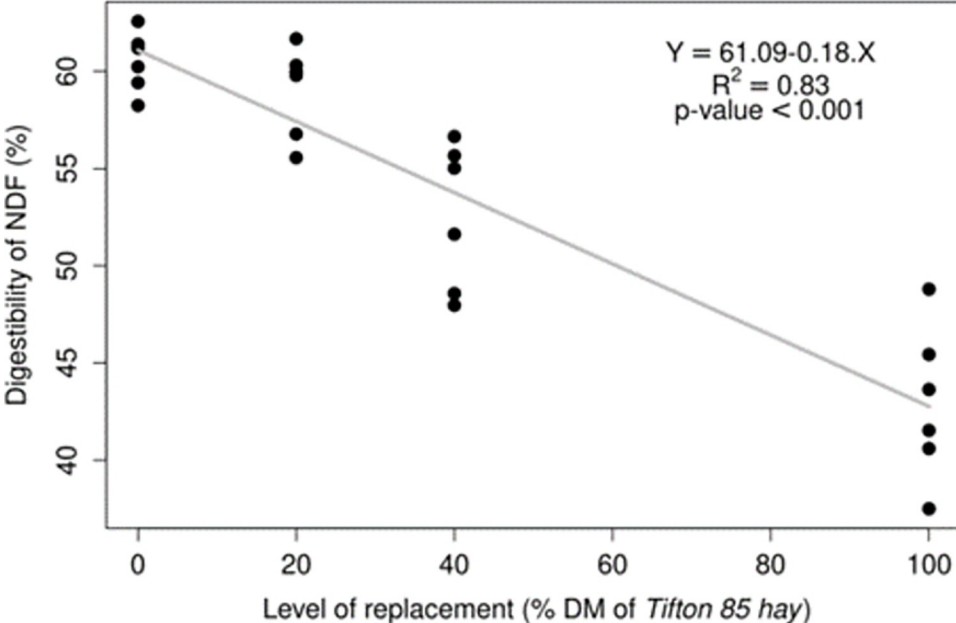

**Fig 2. Digestibility of neutral detergent fiber as a function of Tifton 85 hay (% DM) replacement levels in the diet.**

regression equation, the replacement levels of 20, 40, and 100% would cause a decrease on NDF digestibility by 57.42, 53.76, and 42.77%, respectively. Although, this behavior did not negatively influence lamb performance, an accentuated decrease in NDF digestibility was observed at 100% replacement of Tifton 85 with the Cratylia. Thus, maintaining performance throughout the experiment.

### Effects of Cratylia hay on nitrogen variables

The level of Tifton 85 replacement for Cratylia in the roughage resulted in a decrease on N intake, fecal N, absorbed N, and Retained N (P<0.001) (Table 6). On the other hand, the Urinary N decreased at greater replacement levels.

As far as the collection period, there was a greater (P<0.001) proportion of N intake, fecal N, absorbed N, and urinary N in the final collection period. On the other hand, no difference was observed in the Retained N between the collection periods.

The level of Tifton 85 replacement for Cratylia also resulted in increased proportion (%) of Retained to Absorbed N and Retained N to N Intake (P<0.001). The proportion (%) of Retained to Absorbed N and Retained N to N Intake was greater (P<0.001) in the initial period.

## Discussion

### Lambs' performance on *Cratylia argentea* hay

In most of the measured parameters, the evaluated lambs performed similarly under diets with Tifton 85 replacement for Cratylia hay. Therefore, revealing the capability of Cratylia to be used as animal feed. The use of alternative feedstuff has been a common practice to decrease production cost on livestock systems [27]. Producers without the necessary inputs to grow or purchase Tifton 85 hay, may have an alternative with Cratylia, which is a sustainable feedstuff, since it does not require the use of chemical N fertilizers for high yield and quality biomass production [16].

For the variables of DMI, dry matter digestibility (DMD), FC and FE, Cratylia and Tifton 85 performed the same as well. Dry matter intake was not related to differences in digestibility between the two forage species since the dietary replacement of Tifton 85 for Cratylia did not increase DMD. The particle size uniformity of the diet supplied to the lambs did not allow

**Table 6. Nitrogen balance in feedlot Lacaune lambs fed diets containing different levels of *C. argentea* hay replacing Tifton 85 hay at two evaluation times.**

| | Collection Period | | SE | *P*-value | Level | | | | SE | *P*-value |
|---|---|---|---|---|---|---|---|---|---|---|
| | Initial | Final | | | 0% | 20% | 40% | 100% | | |
| | g/day | | | | | | | | | |
| N intake | 42.14b | 49.63a | 3.71 | <0.001 | 42.88 | 47.82 | 44.99 | 47.86 | 4.14 | <0.001 |
| Fecal N | 11.87b | 13.50a | 0.88 | <0.001 | 11.49b | 12.11b | 12.07b | 13.71a | 1.09 | <0.001 |
| Absorbed N | 30.97b | 36.13a | 2.83 | <0.001 | 31.36b | 35.72a | 32.97ab | 34.15ab | 3.07 | <0.001 |
| Urinary N | 8.69b | 13.68a | 1.46 | <0.001 | 12.17a | 11.45a | 11.24ab | 9.89b | 1.55 | <0.001 |
| Retained N | 22.45 | 22.46 | 1.07 | 0.982 | 19.25b | 23.69a | 23.59a | 23.62a | 1.85 | <0.001 |
| | Retention ratio (%) | | | | | | | | | |
| Retained N/Absorbed N | 72.09a | 62.01b | 1.97 | <0.001 | 61.76c | 68.64b | 66.54b | 71.27a | 2.67 | <0.001 |
| Retained N/N intake | 53.58a | 46.49b | 2.08 | <0.001 | 45.67b | 51.37a | 52.37a | 50.73a | 2.71 | <0.001 |

Means followed by different letters on the same line differ statistically using the Tukey test (p<0.05); SE = standard error; L = level of Tifton 85 replacement levels for *C. argentea* hay in the roughage to replace Tifton 85 hay.

them to select ingredients, which explains the absence of differences in DMI. Thornton [28] reported that the voluntary intake of tropical legumes (when not limited by anti-nutritional factors) is generally greater than that of equally digestible grasses, which may be a consequence of a shorter retention time of legumes in the digestive tract. Likewise, feed particle size may have also influenced the response of the DM digestibility coefficient. The effectiveness of fiber was broken by the particle size of the roughage, so the intestinal passage rate could not reveal possible interactions between the roughages used (Tifton 85 or Cratylia hay). The digestibility coefficient values found in this study are similar to those described by other authors [29], which included shrub hay and tree legume species as roughage sources in the total diet of feed-lot lambs.

The greater DMI with weeks of hay feeding present in this study might be related to an increase in energy required for maintenance, compared to earlier periods of lower BW [30]. Cabral [31] have also found increased DMI and WG in animals with increased weight when evaluating confined lambs. The increased WG supports the findings that Cratylia did not negatively affect the performance on evaluated lambs. The addition of Cratylia on animal feeds was previously reported to increase ($P<0.05$) feed intake from 6.6 to 7.8 and 8.7 kg DM/day, when creole dairy cows were fed sorghum silage alone and supplementation with 2 and 3 kg DM of Cratylia, respectively [32].

## Digestibility

In the matter of OM, CP and NDF digestibility, the increased levels of Cratylia resulted in lower digestibility. Correa Pinzón [17] also reported lower CP and NDF digestibility coefficients in Cratylia hay compared with whole-plant silage or meal. The greater NDF and acid detergent fiber (ADF) contents and lower digestibility of sun-dried forages are attributed to the negative effects of ultraviolet radiation, which can induce the Maillard reaction and thereby increase fiber-bound protein [22,33]. As an example, nitrogen contents bound to the plant cell wall, which become unavailable to the rumen microbiota [34]. Moreover, the plants used for hay production were harvested for high DM production (90 days of regrowth) and cut as a whole, including stems and senescent leaves. Therefore, there was no selection for the less lignified material, which may have contributed to greater indigestible fiber content in the diet.

The NDF content of the plants at the time of harvest in this study was similar to the 67.54% and 65.68% found by [35] in Cratylia plants harvested at 90 days in the rainy and dry seasons, respectively. It is important to highlight that the lignin present in shrub and tree legumes, as in the case of Cratylia, may be less degradable in the rumen compared with the lignin of grasses [34]. However, this knowledge is derived from reports of studies with temperate forages, so the differential action of lignin in tropical legumes is yet to be validated [5].

## Feed conversion and feed efficiency

The greater FC and lower FE in the final period might be attributed to the fact that lambs on the last seven days before finishing were heavier and had a lower growing factor, which limits FE. [36] suggest that older lambs in the finishing phase tend to have different nutrient demands and composition of weight gain (e.g., more fat than muscle gain), thus, lower FE. Polli [37] evaluated finishing lambs in lower and higher heat conditions and found a FE decrease of 12.8% and 20.4% on feedlot lambs as the finishing period progressed, respectively. Bowen [38] implied that FC might be associated with several factors including feed intake, increased age and liveweight of the animal, which explains the findings of this research.

### Nitrogen variables

The increase in N variables at the final collection period (N intake, Fecal N, Absorbed N, Urinary N), might be justified by the greater N content on Cratylia than Tifton 85 (Table 2). Valles-de la Mora [13] described greater urinary N excretion in lambs fed diets supplemented with dry leaves of Cratylia, in comparison with the group fed a basal diet with hay of *Urochloa dictyoneura* cv. Lanero. The authors relate this result to the increased urine volume and intake of available water. A similar behavior was observed in the present experiment, which N excretion was greater in the urine than in the feces, especially in the heavier lambs.

The greater proportion (%) of Retained to Absorbed N and Retained N to N Intake in the initial period possibly happened due to a greater demand of the growing lambs for protein. Since part of the N further is synthesized into proteins by rumen microorganisms [39] to supply the protein demand and increase muscle growth [40,41]. On the other hand, the decrease on Retained to Absorbed N and Retained N to N Intake at the final period could be an indicative of excessive protein in the diet. These results demonstrate an opportunity for considering lower levels of substitution, since part of the N was not being absorbed. Another alternative would be a decrease on external sources of nitrogen (concentrate supplements).

## Conclusion

Overall, the replacement of Tifton 85 for Cratylia hay tested demonstrated that lamb performance can be maintained, regardless of replacement level used in the diets. Nonetheless, a special attention is needed when looking at 100% substitution level because a few parameters such as the digestibility of NDF, OM and CP were decreased. Furthermore, the N retention was lower, which was excreted into the soil.

Thus, might be recommended as a food intervention strategy to support livestock production at no cost of chemical fertilizers, since none was used for Cratylia hay production. Future research may evaluate lower levels of protein concentrates associated with *C. argentea* and the economic viability of this feed, compared to other roughage.

## Supporting information

**S1 File.**
(XLSX)

## Author Contributions

**Conceptualization:** Elaine Cristina Teixeira, Luciano Soares de Lima, Hemilly Cristina Menezes de Sa, Ângela Maria Quintão Lana.

**Data curation:** Elaine Cristina Teixeira, Fernando Antônio de Souza.

**Formal analysis:** Fernando Antônio de Souza.

**Funding acquisition:** Ângela Maria Quintão Lana.

**Investigation:** Elaine Cristina Teixeira, Lucas Freires Abreu.

**Methodology:** Elaine Cristina Teixeira, Luciano Soares de Lima, Hemilly Cristina Menezes de Sa.

**Project administration:** Elaine Cristina Teixeira.

**Resources:** Elaine Cristina Teixeira, Walter José Rodrigues Matrangolo, Karina Toledo da Silva, Ângela Maria Quintão Lana.

**Software:** Fernando Antônio de Souza.

**Supervision:** Elaine Cristina Teixeira, Walter José Rodrigues Matrangolo, Karina Toledo da Silva, Ângela Maria Quintão Lana.

**Validation:** Elaine Cristina Teixeira.

**Visualization:** Elaine Cristina Teixeira.

**Writing – original draft:** Elaine Cristina Teixeira.

**Writing – review & editing:** Elaine Cristina Teixeira, Lucas Freires Abreu, Ângela Maria Quintão Lana.

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
