## [Decision Letter · Decision Letter 0]

12 Mar 2023

PONE-D-23-03057Could Cratylia argentea replace Tifton 85 hay on finishing lamb diets in tropical areas?PLOS ONE

Dear Dr. Elaine,

Thank you for submitting your manuscript to PLOS ONE. After careful consideration, we feel that it has merit but does not fully meet PLOS ONE’s publication criteria as it currently stands. Therefore, we invite you to submit a revised version of the manuscript that addresses the points raised during the review process.

We look forward to receiving your revised manuscript.

Kind regards,

Marcia Saladini Vieira Salles

Academic Editor

PLOS ONE

Journal Requirements:

4. Thank you for stating the following in the Financial Disclosure Statement of your manuscript: 

"The funders had no role in study design, data collection and analysis, decision to publish, or preparation of the manuscript. This work was supported by the National Council of Technological and Scientific Development (CNPq, Brazil #140314/2019-9), Minas Gerais State Agency for Research and Development (FAPEMIG, PPM-00639-18) and the Coordination for the Improvement of Higher Education Personnel (CAPES, Brazil #88887.667823/2022-00). "

"This work was supported by the National Council for Scientific and Technological Development (CNPq, Brazil nº 140314/2019-9), State Agency for Research and Development of Minas Gerais (FAPEMIG, PPM-00639-18) and Coordination for the Improvement of Higher Education Personnel of Education (CAPES, Brazil nº 88887.667823/2022-00).

Additional Editor Comments:

Summary: describe in the summary the total period of the experiment. Weight gain between treatments was not shown in the results.

L70 to 73: Please insert in the sentence the justification for why it would be more suitable for small production systems.

The introduction is too long.

Suggestion: text from L108 to 123 can be used in the discussion and removed from the introduction.

Authors need to be careful in claiming that Chrysalia hay can replace 100 % Tifton hay due to the value of nutrients found in Tifton hay that was used in the experiment.

M&M: Why do a digestibility test at the beginning and another at the end?

L 150: Could you please verify that this FDN value is correct? It seems very low to me than normally found for this food.

Table 3: Data can be described in the text. There is no need for them to be transcribed into a table.

L283 to 285: I think the authors are misinterpreting the results. In the way the results were arranged, when there is the significance of L (treatment) as only one P was generated, it means that the results used for the statistics were the data set of both periods (715.1 and 716.2) which would be an average close to 715.65 for 0 and 690.9 for 100% treatment (681.7 and 700.1). If the P is specific for the initial period, then the authors must provide the other probabilities and indicate with letters where there was a difference.

L290 to 292: display different letters when there is a statistical difference between means. The table should be self-explanatory and show all existing results.

Table 6. Data are hard to follow. It would be interesting for the authors to provide a more didactic disposition of the results.

Tables. Present two decimal places after the point to better understand the variability of the data. The standard error of the mean is two decimal places.

Statistics: Did the authors test the strength of the experiment? From the averages found in most of the variables and the respective standard errors, it is doubtful that there was no difference because the experiment was not strong enough to obtain differences due to the low repetition of animals per treatment.

L318 to 320. The authors describe that the animals were weighed weekly and do not present any weight or weight gain data from the treatments. Only in time. Please present the results with the respective standard errors of the mean and probabilities. It is also important to better visualize the efficiency results.

Results: The authors have the digestibility coefficients and did not present the nutrient intake. Why? If we do the DMI calculations, animals with 100 g of C. argentea hay ingested 300 g more food per day compared to 100 g of Tifton. Is it worth it, isn't it worth it for food costs? This would be an interesting part for the authors to discuss in the 'Discussion' section.

Discussion: The discussion needs to be revised and improved. In most of the text, the authors discuss the differences in time due to the growth of the animals and compare them with the results of other authors. They do not bring a discussion that advances knowledge about the purpose of the study.

The authors affirm the high content of lignin in the hay of C. argentea but do not present the results of this component in the work. I think that to use this discussion it would be interesting to present the results of lignin and/or to present average data on the amount of this component of other authors who used this food in their research.

L406: The fact that the greater the amount of ingested protein causes more water consumption is explained by physiology. To excrete nitrogen the organism needs to do it together with water. Please review physiology studies.

Conclusion: The authors need to reflect a little more on the results obtained to conclude the complete replacement of C. argentea hay by Tifton hay, given the high variability of the results and the low repetition of experimental units. Mainly by the results of digestibility coefficients found. Unless even the food has worse digestibility, and with the additional cost of offering this food to obtain the same weight gain, it is economically compensated for the producer.

Conclusion: Please remove the last paragraph from the conclusion because nutrient parameters in different hays harvested at different times were not studied.

Please get the references right.

Reviewers' comments:

Reviewer's Responses to Questions

**Comments to the Author**

1. Is the manuscript technically sound, and do the data support the conclusions?

Reviewer #1: Yes

Reviewer #2: Yes

2. Has the statistical analysis been performed appropriately and rigorously? 

Reviewer #1: Yes

Reviewer #2: Yes

3. Have the authors made all data underlying the findings in their manuscript fully available?

Reviewer #1: Yes

Reviewer #2: Yes

4. Is the manuscript presented in an intelligible fashion and written in standard English?

Reviewer #1: Yes

Reviewer #2: Yes

5. Review Comments to the Author

Reviewer #1: Could Cratylia argentea replace Tifton 85 hay on finishing lamb diets in tropical areas? it is an original work and brings a good proposal for an alternative replacement to a food that is already consolidated in the diet of ruminants in Brazil. The work is adequate and offers sufficient subsidies for publication in Plos One, provided that the authors consider all the suggestions made in the manuscript. The title begins with a question, based on the findings, the authors are expected to answer in the conclusion clearly and to recommend the replacement level.

Reviewer #2: General points about the manuscript: The manuscript brings interesting information regarding evaluation of alternative feedstuff. The manuscript is very well vritten, with sufficicient information, but there are some especifics issues that must be reviewed.

Line 142 to 148: The paragraph has significant wordiness and awkward phraseology. Please try to make the paragraph more explanatory. Rewriter. It is confusing.

Line 174 – Drums is the best word to use? Check it.

Line 176 - …November 2020 and lasted January 2020. Is the year “January 2020” correct?

The lambs stay 64 days in an individual metabolic cages?

The word “orts” is the best word to the context?

6. PLOS authors have the option to publish the peer review history of their article (what does this mean?). If published, this will include your full peer review and any attached files.

Reviewer #1: No

Reviewer #2: No

---

## [Author Response · Author response to Decision Letter 0]

25 Sep 2023

Response to Reviewers

Journal Requirements:

A: adjusted.

A: adjusted.

A: adjusted. Information over the “cover letter” file.

4. Thank you for stating the following in the Financial Disclosure Statement of your manuscript: 

"The funders had no role in study design, data collection and analysis, decision to publish, or preparation of the manuscript. This work was supported by the National Council of Technological and Scientific Development (CNPq, Brazil #140314/2019-9), Minas Gerais State Agency for Research and Development (FAPEMIG, PPM-00639-18) and the Coordination for the Improvement of Higher Education Personnel (CAPES, Brazil #88887.667823/2022-00). "

"This work was supported by the National Council for Scientific and Technological Development (CNPq, Brazil nº 140314/2019-9), State Agency for Research and Development of Minas Gerais (FAPEMIG, PPM-00639-18) and Coordination for the Improvement of Higher Education Personnel of Education (CAPES, Brazil nº 88887.667823/2022-00). The funders had no role in study design, data collection and analysis, decision to publish, or preparation of the manuscript."

A: Reviewer, the amended information was moved to the Cover Letter, as requested.

A: Information adjusted, as requested.

Additional Editor Comments:

Summary: describe in the summary the total period of the experiment. Weight gain between treatments was not shown in the results.

Figure 1A answers this. From the graph, it can be seen that the animals gained weight at a rate of 1.93 kg per week, which gives an average daily gain of (1.93/7 = 275 grams/day).

L70 to 73: Please insert in the sentence the justification for why it would be more suitable for small production systems.

Authors decided to remove this sentence.

The introduction is too long.

Adjusted, as requested.

Suggestion: text from L108 to 123 can be used in the discussion and removed from the introduction.

Indeed. Thanks for the suggestion. 

Authors need to be careful in claiming that Cratylia hay can replace 100 % Tifton hay due to the value of nutrients found in Tifton hay that was used in the experiment.

M&M: Why do a digestibility test at the beginning and another at the end?

As this is a new food, we sought to evaluate the performance of the animals at different times during the experiment. That is, growing animals and animals already finishing.

L 150: Could you please verify that this FDN value is correct? It seems very low to me than normally found for this food.

The Neutral Fiber Detergent (NFD) values are consistent with those found in other studies.

Gama, T. D. C. M., Zago, V. C. P. Nicodemo, M. L. F., Laura, V. A., Volpe, E., & Morais, D.G. (2009). Composição bromatológica, digestibilidade in vitro e produção de biomassa de leguminosas forrageiras lenhosas cultivadas em solo arenoso. Revista Brasileira de Saúde e Produção Animal, 10(3), 560-572.

Montero-Durán, E., Rojas-Bourrillon, A., & López-Herrera, M. (2021). Sustitución de Cratylia argentea y Erythrina poeppigiana por guineo cuadrado en ensilados. Nutrición Animal Tropical, 15(2), 123-146. https://doi.org/10.15517/nat.v15i2.48818

ÁLVAREZ CARRILLO, F.; FERNANDO CASANOVES; CUELLAR MEDINA, Y.; ORTIZ MENESES, J. F.; BALANTA MARTINEZ, V. J.; CELIS PARRA, G. A. Nutritional quality of Piptocoma discolor and Cratylia argentea as a non-timber forest products for animal feed in the Caquetá province. Journal of Agriculture and Environment for International Development (JAEID), [S. l.], v. 116, n. 2, p. 109–120, 2023. DOI: 10.36253/jaeid-13102. https://www.jaeid.it/index.php/jaeid/article/view/13102

Table 3: Data can be described in the text. There is no need for them to be transcribed into a table.

Adjusted.

L283 to 285: I think the authors are misinterpreting the results. In the way the results were arranged, when there is the significance of L (treatment) as only one P was generated, it means that the results used for the statistics were the data set of both periods (715.1 and 716.2) which would be an average close to 715.65 for 0 and 690.9 for 100% treatment (681.7 and 700.1). If the P is specific for the initial period, then the authors must provide the other probabilities and indicate with letters where there was a difference.

L290 to 292: display different letters when there is a statistical difference between means. The table should be self-explanatory and show all existing results.

Table 5. Data are hard to follow. It would be interesting for the authors to provide a more didactic disposition of the results.

Tables. Present two decimal places after the point to better understand the variability of the data. The standard error of the mean is two decimal places.

Adjusted.

Statistics: Did the authors test the strength of the experiment? From the averages found in most of the variables and the respective standard errors, it is doubtful that there was no difference because the experiment was not strong enough to obtain differences due to the low repetition of animals per treatment.

To detect small differences between the means, for data that have a high standard deviation, a high number of repetitions would be necessary, that is, a very large number of animals, which could make timely research unfeasible, as approved by the ethics council.

L318 to 320. The authors describe that the animals were weighed weekly and do not present any weight or weight gain data from the treatments. Only in time. Please present the results with the respective standard errors of the mean and probabilities. It is also important to better visualize the efficiency results.

Added table with this information

Results: The authors have the digestibility coefficients and did not present the nutrient intake. Why? If we do the DMI calculations, animals with 100 g of C. argentea hay ingested 300 g more food per day compared to 100 g of Tifton. Is it worth it, isn't it worth it for food costs? This would be an interesting part for the authors to discuss in the 'Discussion' section.

Indeed. Thanks for the suggestion. 

The objective of the present study is to evaluate Cratylia as a new food that can be introduced into the market for feeding ruminants, however, there is no pricing yet. However, the productive seasonality of tropical forage grasses, such as Tifton-85 grass, requires investments in infrastructure and machinery, which consequently reduce the net profitability of the production system. Therefore, the use of Cratylia can be an interesting alternative, especially where conditions for the cultivation of other forage species may be limited.

Discussion: The discussion needs to be revised and improved. In most of the text, the authors discuss the differences in time due to the growth of the animals and compare them with the results of other authors. They do not bring a discussion that advances knowledge about the purpose of the study.

The authors affirm the high content of lignin in the hay of C. argentea but do not present the results of this component in the work. I think that to use this discussion it would be interesting to present the results of lignin and/or to present average data on the amount of this component of other authors who used this food in their research.

Adjusted.

L406: The fact that the greater the amount of ingested protein causes more water consumption is explained by physiology. To excrete nitrogen the organism needs to do it together with water. Please review physiology studies.

Indeed. Thanks for the suggestion. 

Conclusion: The authors need to reflect a little more on the results obtained to conclude the complete replacement of C. argentea hay by Tifton hay, given the high variability of the results and the low repetition of experimental units. Mainly by the results of digestibility coefficients found. Unless even the food has worse digestibility, and with the additional cost of offering this food to obtain the same weight gain, it is economically compensated for the producer.

Conclusion: Please remove the last paragraph from the conclusion because nutrient parameters in different hays harvested at different times were not studied.

Adjusted.

Please get the references right.

Reviewers' comments:

Reviewer's Responses to Questions

Comments to the Author

1. Is the manuscript technically sound, and do the data support the conclusions?

Reviewer #1: Yes

Reviewer #2: Yes

2. Has the statistical analysis been performed appropriately and rigorously?

Reviewer #1: Yes

Reviewer #2: Yes

3. Have the authors made all data underlying the findings in their manuscript fully available?

Reviewer #1: Yes

Reviewer #2: Yes

4. Is the manuscript presented in an intelligible fashion and written in standard English?

Reviewer #1: Yes

Reviewer #2: Yes

5. Review Comments to the Author

Reviewer #1: “Could Cratylia argentea replace Tifton 85 hay on finishing lamb diets in tropical areas?” it is an original work and brings a good proposal for an alternative replacement to a food that is already consolidated in the diet of ruminants in Brazil. The work is adequate and offers sufficient subsidies for publication in Plos One, provided that the authors consider all the suggestions made in the manuscript. The title begins with a question, based on the findings, the authors are expected to answer in the conclusion clearly and to recommend the replacement level.

Adjusted in the authors' best discretion

Reviewer #2: General points about the manuscript: The manuscript brings interesting information regarding evaluation of alternative feedstuff. The manuscript is very well vritten, with sufficicient information, but there are some especifics issues that must be reviewed.

Line 142 to 148: The paragraph has significant wordiness and awkward phraseology. Please try to make the paragraph more explanatory. Rewriter. It is confusing.

Adjusted in the authors' best discretion.

Line 174 – Drums is the best word to use? Check it. 

Adjusted.

Line 176 - …November 2020 and lasted January 2020. Is the year “January 2020” correct? 

Adjusted.

The lambs stay 64 days in an individual metabolic cages?

Yes.

The word “orts” is the best word to the context?

Yes, it is.

---

## [Decision Letter · Decision Letter 1]

30 Oct 2023

PONE-D-23-03057R1Could Cratylia argentea replace Tifton 85 hay on finishing lamb diets in tropical areas?PLOS ONE

Dear Dr. Teixeira,

Thank you for submitting your manuscript to PLOS ONE. After careful consideration, we feel that it has merit but does not fully meet PLOS ONE’s publication criteria as it currently stands. Therefore, we invite you to submit a revised version of the manuscript that addresses the points raised during the review process.

 Please submit your revised manuscript by November 20th. If you will need more time than this to complete your revisions, please reply to this message or contact the journal office at plosone@plos.org. Please include the following items when submitting your revised manuscript:A rebuttal letter that responds to each point raised by the academic editor and reviewer(s). You should upload this letter as a separate file labeled 'Response to Reviewers'.A marked-up copy of your manuscript that highlights changes made to the original version. You should upload this as a separate file labeled 'Revised Manuscript with Track Changes'.An unmarked version of your revised paper without tracked changes. You should upload this as a separate file labeled 'Manuscript'.If applicable, we recommend that you deposit your laboratory protocols in protocols.io to enhance the reproducibility of your results. Protocols.io assigns your protocol its own identifier (DOI) so that it can be cited independently in the future. For instructions see: https://journals.plos.org/plosone/s/submission-guidelines#loc-laboratory-protocols. Additionally, PLOS ONE offers an option for publishing peer-reviewed Lab Protocol articles, which describe protocols hosted on protocols.io. Read more information on sharing protocols at https://plos.org/protocols?utm_medium=editorial-email&utm_source=authorletters&utm_campaign=protocols.

We look forward to receiving your revised manuscript.

Kind regards,

Marcia Saladini Vieira Salles

Academic Editor

PLOS ONE

Journal Requirements:

Additional Editor Comments:

The authors sent the first version without line numbers, which made the manuscript review process very difficult. Please insert line numbers in the next revision.

Be careful when sending files. They include comments from the authors.

Please correct the punctuation in the introduction. By changing the names of the authors to numbers, the text was distorted. There are also other parts of the text that need to be revised for spacing and punctuation.

Abstract: The objective of this study was to evaluate the performance of 'growing' and finishing Lacaune lambs fed Cratylia argentea hay as an alternative to Tifton 85 (Cynodon spp).

Keywords suggestion: forage conservation, forage digestibility, ruminant feeding, shrub legume, small ruminant performance

In the Results section, please present the real probabilities found for the respective variables and not less than 0.05.

Please provide ADF values for Tifton hay.

It was requested that the lignin values of the respective hays be presented, mainly from Cratylia argentea as there was discussion about this component, the authors responded that it was adjusted but I did not find it in the final version of the manuscript.

Table 5: if there was no interaction between treatment and period, please remove the P values for LxC.

Table 5 = table 6?????

Table 7: Please correct. The value presented as BW does not match the weight of the animals presented at the beginning of the M&M of 21.5 ± 3.38 kg. I think the authors must be presenting weight gain here. But I ask the authors' attention because in the answer given to the questions it was written that the weight gain was 275g/day and the table says 250 g/day.Favor verificar os dados apresentados na tabela 6 e 7.

Table 6 0% 20% 40% 100%

DMI 1.50 1.58 1.47 1.64

Digestibility 692.72 697.54 688.64 672.84

" Actual MS intake 1.035 1.09 0.99 1.10"

Table 7 1.50 1.60 1.51 1.66

There are two tables 7. In the second table 7, as there was no interaction between LxC, please remove it from the table and only describe in the text that there was no interaction for these variables.

Please present this nitrogen data in the same format shown in table 6.

Discussion: the 100% C. argentea treatment had the worst N retention ratio. This means greater nitrogen excretion into the soil. And this is not desirable. I think the authors should take this into account in the discussion and conclusion.

Conclusion: digestibility results were worse for C. argentea hay when at 100% replacement. We can see that numerically the conversion efficiency values were worse for this treatment (100%). No difference was detected because variability is high. Absence of evidence is not evidence of absence. I suggest the authors evaluate their conclusion in suggesting that C. argentea hay can replace Tifton hay by 100%.

General suggestions: consider using a greater number of repetitions per treatment in future experiments. Six is a low number for performance experiments due to the variability of the respective variable.

Reviewers' comments:

Reviewer's Responses to Questions

**Comments to the Author**

1. If the authors have adequately addressed your comments raised in a previous round of review and you feel that this manuscript is now acceptable for publication, you may indicate that here to bypass the “Comments to the Author” section, enter your conflict of interest statement in the “Confidential to Editor” section, and submit your "Accept" recommendation.

Reviewer #1: All comments have been addressed

Reviewer #2: All comments have been addressed

2. Is the manuscript technically sound, and do the data support the conclusions?

Reviewer #1: Yes

Reviewer #2: Yes

3. Has the statistical analysis been performed appropriately and rigorously? 

Reviewer #1: Yes

Reviewer #2: Yes

4. Have the authors made all data underlying the findings in their manuscript fully available?

Reviewer #1: Yes

Reviewer #2: Yes

5. Is the manuscript presented in an intelligible fashion and written in standard English?

Reviewer #1: Yes

Reviewer #2: Yes

6. Review Comments to the Author

Reviewer #1: (No Response)

Reviewer #2: (No Response)

7. PLOS authors have the option to publish the peer review history of their article (what does this mean?). If published, this will include your full peer review and any attached files.

Reviewer #1: No

Reviewer #2: No

---

## [Author Response · Author response to Decision Letter 1]

9 Nov 2023

Information attached to the submisstion and renamed as "Response to Reviewers 2 - PONE-D-23-03057 - 2023-11-07".

---

## [Editor Report · Decision Letter 2]

13 Nov 2023

PONE-D-23-03057R2Could Cratylia argentea replace Tifton 85 hay on growing and finishing lamb diets in tropical areas?PLOS ONE

Dear Dr. Teixeira,

Thank you for submitting your manuscript to PLOS ONE. After careful consideration, we feel that it has merit but does not fully meet PLOS ONE’s publication criteria as it currently stands. Therefore, we invite you to submit a revised version of the manuscript that addresses the points raised during the review process.

Please submit your revised manuscript by Dec 28 2023 11:59PM. If you will need more time than this to complete your revisions, please reply to this message or contact the journal office at plosone@plos.org. Please include the following items when submitting your revised manuscript:A rebuttal letter that responds to each point raised by the academic editor and reviewer(s). You should upload this letter as a separate file labeled 'Response to Reviewers'.A marked-up copy of your manuscript that highlights changes made to the original version. You should upload this as a separate file labeled 'Revised Manuscript with Track Changes'.An unmarked version of your revised paper without tracked changes. You should upload this as a separate file labeled 'Manuscript'.If applicable, we recommend that you deposit your laboratory protocols in protocols.io to enhance the reproducibility of your results. Protocols.io assigns your protocol its own identifier (DOI) so that it can be cited independently in the future. For instructions see: https://journals.plos.org/plosone/s/submission-guidelines#loc-laboratory-protocols. Additionally, PLOS ONE offers an option for publishing peer-reviewed Lab Protocol articles, which describe protocols hosted on protocols.io. Read more information on sharing protocols at https://plos.org/protocols?utm_medium=editorial-email&utm_source=authorletters&utm_campaign=protocols.

We look forward to receiving your revised manuscript.

Kind regards,

Marcia Saladini Vieira Salles

Academic Editor

PLOS ONE

Journal Requirements:

**Additional Editor Comments:**

Dear Dr. Elaine,

Thank you for resubmitting your manuscript to PLOS ONE.

I have completed the review of your manuscript and a summary is appended below.

There are only minor adjustments to the manuscript.

Please, when sending the revised manuscript again, send it with the accepted considerations highlighted in yellow and not in the 'control changes' review mode. This makes the review work very difficult.

Please present this nitrogen data (table 6) in the same format shown in table 5. It was requested in the previous opinion.

In the nitrogen table 6 there is a number marked (3) as excluded which I think was not meant to be excluded (13.83). Please check.

---

## [Editor Report · Decision Letter 3]

23 Nov 2023

Could Cratylia argentea replace Tifton 85 hay on growing and finishing lamb diets in tropical areas?

PONE-D-23-03057R3

Dear Dr. Teixeira,

We’re pleased to inform you that your manuscript has been judged scientifically suitable for publication and will be formally accepted for publication once it meets all outstanding technical requirements.

Kind regards,

Marcia Saladini Vieira Salles

Academic Editor

PLOS ONE

---

## [Editor Report · Acceptance letter]

1 Dec 2023

PONE-D-23-03057R3 

Could *Cratylia argentea* replace Tifton 85 hay on growing and finishing lamb diets in tropical areas? 

Dear Dr. Teixeira:

I'm pleased to inform you that your manuscript has been deemed suitable for publication in PLOS ONE. Congratulations! Your manuscript is now with our production department. 

Kind regards, 

on behalf of

Dr. Marcia Saladini Vieira Salles 

Academic Editor

PLOS ONE